# Long-Term Risk Factor Control After Myocardial Infarction—A Need for Better Prevention Programmes

**DOI:** 10.3390/jcm8081114

**Published:** 2019-07-27

**Authors:** Rico Osteresch, Andreas Fach, Johannes Schmucker, Ingo Eitel, Harald Langer, Rainer Hambrecht, Harm Wienbergen

**Affiliations:** 1Bremer Institute for Heart and Circulation Research at the Klinikum Links der Weser, 28277 Bremen, Germany; 2University Heart Center, 23538 Lübeck, Germany

**Keywords:** myocardial infarction, risk factor control, socially disadvantaged districts, prevention programmes

## Abstract

Introduction: Long-term prognosis of myocardial infarction (MI) is still serious, especially in patients with MI and cardiogenic shock. To improve long-term prognosis and prevent recurrent events, sustainable cardiovascular risk factor control (RFC) after MI is crucial. Methods: The article gives an overview on health care data regarding RFC after MI and presents recent trials on modern preventive strategies that support patients to achieve risk factor targets during long-term course. Results: International registry studies, such as EUROASPIRE, observed alarming deficiencies in RFC after MI. As data of the German Bremen ST-segment elevation myocardial infarction (STEMI)-Registry show, most deficiencies are found in socially disadvantaged city districts and in young patients. Several studies on prevention programmes to improve RFC after MI reported inconsistent data; however, in the recently published IPP trial a 12-months intensive prevention programme that included both repetitive personal contacts with non-physician prevention assistants and telemetric risk factor control, was associated with significant improvements of numerous risk factors (smoking, LDL and total cholesterol, systolic blood pressure and physical inactivity). Conclusions: There is a strong need of action to improve long-term risk RFC after MI, especially in socially disadvantaged patients. Modern prevention programmes, using personal and telemetric contacts, have large potential to support patients in achieving long-term risk factor targets after coronary events.

## 1. Introduction: Long-Term Prognosis after Myocardial Infarction

Despite decreasing mortality rates in the last decades, long-term prognosis after myocardial infarction (MI) is still serious [1,2,3].

In a long-term outcome study of Swedish national registries, investigating 97,254 patients admitted with MI and alive one week after discharge, Jernberg et al. reported that 18.3% of the patients suffered from the combined endpoint of cardiovascular death, non-fatal MI or non-fatal stroke during the following 365 days [2]. Additionally, 20% of the patients without a cardiovascular event during the first 365 days experienced a clinical event during the next 36 months [2].

In a five-year long-term analysis of the German Bremen STEMI-Registry (BSR), including 3736 interventionally treated patients with STEMI, survival rates of 87.7% after one year and 78.7% after five years were found [4]. The highest mortality was observed in patients with cardiogenic shock complicating acute STEMI with an in-hospital mortality rate of 38% and a one-year mortality rate of 50%, with a decrease of mortality rates during the period from 2006 to 2013 [3].

It is therefore obvious that management of STEMI-patients with cardiogenic shock (e.g., revascularization therapies, intensive care unit treatment and assist devices) is one crucial challenge for mortality reduction after MI [5].

A further important challenge is the optimization of long-term risk factor control (RFC) after hospital discharge. It has been shown by different studies that sufficient control of cardiovascular risk factors is associated with improved long-term prognosis [6,7,8]. Large subanalyses of the Organization to assess strategies in acute ischemic syndromes (OASIS) 5 trial [6] and the Clinical outcomes utilizing revascularization and aggressive drug evaluation (COURAGE) trial [7] demonstrated significant reductions of major cardiovascular adverse events in patients who reached risk factor goals—even after adjustment for potentially confounding prognostic factors in multivariate analyses.

The present article gives an overview on recent health care data regarding RFC in patients after MI and presents data on trials investigating modern preventive strategies to achieve risk factor targets during long-term course.

Besides own research, a PubMed literature search using the key terms “prevention programme”, “myocardial infarction” and “risk factor control”, as well as checking reference lists of journal articles was performed to identify relevant studies for review.

## 2. “Real World“ Data on Long-Term RFC in Patients after MI

International health care data observed alarming deficiencies in long-term RFC after MI [9,10,11,12]. The recent European action on secondary and primary prevention by intervention to reduce events (EUROASPIRE) V survey that investigated secondary prevention in 8,261 patients with coronary artery disease (CAD) in 27 countries reported unhealthy lifestyles in terms of smoking, diet and sedentary behaviour, with a worsening of lifestyle risk factors compared to EUROASPIRE IV five years before. In detail, 19% of the CAD-patients were active smokers, 59% were centrally obese, 71% had LDL-cholesterol ≥ 1.8 mmol/l (≥ 70 mg/dl) and 42% had a blood pressure ≥ 140/90 mmHg (≥ 140/85 mmHg if diabetic) [12].

Registry data focusing on lipids and lipid-lowering medication in patients with CAD also reported disappointing data: in the international DYSIS II, trial lipid profile was collected at a physician visit in 6,794 patients with stable CAD or 120 days after hospitalization for acute coronary syndromes in 3,867 patients. LDL cholesterol levels below 70 mg/dl were observed in only 29.4%, and 18.9% of patients, respectively [13].

Health care data of the Bremen STEMI-Registry (BSR) reported that most deficiencies were observed in socially disadvantaged patients and in the young. Schmucker et al. investigated the association of socioeconomic status and incidence as well as clinical course of STEMI, utilizing postal codes of the home address of patients and a standardized social deprivation index in Bremen/Germany [14]. The authors found a negative association between low socioeconomic status and incidence of STEMIs per 100,000 inhabitants per year (Figure 1).

Socioeconomically deprived patients had higher rates of smoking and obesity. Although acute treatment strategies were comparable between the different groups of socioeconomic status, a 34% increase of MACCE was observed during a five-year course after MI in patients from the most socially deprived city areas compared to those from the city areas with the highest socioeconomic status. These results were pronounced in young STEMI-patients [14]. The data underline the importance of socioeconomic status in cardiovascular prevention. In the context of a lifetime approach to prevention—that is emphasized in the current guidelines [8]—intensified prevention efforts in socioeconomic deprived areas are a promising strategy to improve long-term prognosis after MI.

Wienbergen et al. investigated long-term RFC in young patients with STEMI [15]. In 277 patients who experienced STEMI at age of ≤45 years and who were revisited after a period of 5.7 ± 4 years, only a minority achieved risk factor targets at the follow-up visits: 38.3% were active smokers, mean body mass index was 29.9 ± 5.1 kg/m^2^ and just 14.8% had a body mass index < 25.0 kg/m^2^. A majority of patients (66.8%) were physically inactive and performed sports less than three days per week. Mean LDL cholesterol level was 94 ± 38 mg/dl, only 27.1% achieved LDL cholesterol levels < 70 mg/dl [15].

## 3. Preventive Strategies

The disappointing health care data on RFC in CAD-patients suggest that effective long-term prevention programmes are missing. Several studies have shown that short-term cardiac rehabilitation programs after coronary events do not provide sustainable prevention effects [8,16]. Different studies on longer-term prevention programmes have been performed [17,18,19,20,21,22,23]: In the EUROACTION trial (2003–2006) a 16-week nurse-coordinated prevention programme was associated with a trend towards better smoking prevention and lower LDL cholesterol in patients with CAD; however, no relevant effects on total cholesterol or body weight were observed [18]. More current studies showed inconsistent results on prevention programmes [19,20,21,22,23]: in the Optimal cardiac rehabilitation (OPTICARE) trial, long-term preventive contacts after acute coronary syndromes were not superior to standard rehabilitation in reducing the primary endpoint [Systematic coronary risk evaluation (SCORE)] [21,22]. In the Randomized evaluation of secondary prevention by outpatient nurse specialists (RESPONSE) II trial commercial lifestyle programmes (such as “Weight Watchers^®^ (New York City, NY, USA“) led to a reduction of the primary endpoint “prevention success“ (improvement of one risk factor without deterioration of another); this was largely triggered by a significant reduction of body weight [23].

In the most recent IPP trial a 12-months intensive prevention programme (IPP) after MI was compared to usual care (UC) [24]: one month after discharge for MI, mostly (96%) after a 3-week cardiac rehabilitation programme, patients were randomly assigned to the IPP for 12 months or UC. IPP was coordinated by a non-physician prevention assistant that was supervised by cardiologists. The prevention programme included group education sessions every month, personal telephone contacts every three weeks, clinical visits if risk factors did not meet the guideline-recommended targets and telemetric devices with online documentation of risk factors. After 12 months IPP was associated with a significantly better RFC compared to UC with a lower rate of recurrent smokers (3.2% vs. 16.4%, *p* < 0.05), lower levels of LDL cholesterol (67.6 ± 21 mg/dl vs. 78.4 ± 29 mg/dl, *p* < 0.05) and total cholesterol (143.2 ± 26 mg/dl vs. 153.3 ± 34 mg/dl, *p* < 0.05). In the IPP group lower systolic blood pressure levels (130 ± 15 mmHg vs. 135 ± 18 mmHg, *p* < 0.05) and a higher rate of physical activity (157% vs. 12% increase of caloric expenditure, *p* < 0.01) were observed.

To illustrate the effects of IPP the study endpoints “prevention success” and “prevention failure” are shown in Figure 2. These endpoints were developed in analogy to the RESPONSE II trial [23]. Table 1 summarizes the definition of “prevention success” and “prevention failure” used for the IPP study, based on previous studies on cardiovascular prevention [8,23]

Prevention success was observed in 42.8% of the patients in the IPP study arm compared to 29.4% in the UC group (*p* < 0.05). Prevention failure was observed in 22.5% in IPP vs. 38.5% in UC (*p* < 0.05) (Figure 2).

Comparing the RESPONSE II [23] and the IPP [24] trial, both studies investigated a modern long-term prevention programme after acute coronary syndromes and evaluated “prevention success” and “prevention failure” after 12 months. However, in the RESPONSE II trial exclusively lifestyle interventions (such as “Weight Watchers^®^”) were performed and the three “lifestyle risk factors”; body weight, physical inactivity and smoking, were defining “prevention success” or “prevention failure”. In contrary, improvement of medical therapy (besides lifestyle interventions) was an important component of the prevention programme in the IPP trial, e.g., advising patients to increase lipid-lowering medication if LDL cholesterol levels did not meet the recommended targets. Therefore, and due to the better evidence of LDL cholesterol reduction compared to weight reduction to improve prognosis after MI, LDL cholesterol instead of weight modification was used to calculate “prevention success” and “prevention failure” in the IPP study.

All three components of “prevention success“ and “prevention failure“—LDL cholesterol, smoking and physical activity—were significantly improved in the IPP study (while body weight was not significantly reduced). In contrast, in the RESPONSE II study only body weight was significantly reduced, while the risk factors smoking and physical inactivity were not improved (nor was LDL cholesterol).

The comparison of both studies demonstrates that the focus of preventive programmes (e.g., weight reduction or lipid-lowering) has strong implications on achieved prevention results.

Different subanalyses on the prevention effects of IPP have been performed. It was found that patients who used telemetric control of physical activity and increased their numbers of daily step > 30% during study course had a better improvement of risk factors compared to patients with no or small increase of steps numbers (body mass index—3.9% vs. ± 0.0%, *p* < 0.05, systolic blood pressure—4.9% vs. + 1.5%, *p* < 0.05) [25].

In a further subanalysis it was observed that MI-patients with low school-leaving qualifications had more risk factors at time of MI than patients with higher school graduations; however, patients with lower school-leaving qualifications showed a greater improvement of risk factor profile by the long-term prevention programme compared to the patients with higher school graduations [26]. These data support preventive strategies that focus on patients with lower educational or socioeconomic status.

## 4. Discussion: How to Improve Long-Term Risk Factor Control?

Recent registry data have shown that there is a strong need of action to improve long-term risk RFC after MI, especially in socially disadvantaged patients and in young patients. Different trials investigated the effects of longer-term prevention programmes (EUROACTION, OPTICARE, RESPONSE), the most recent IPP trial reported that a 12-months intensive prevention programme, coordinated by non-physician prevention assistants and including telemetric strategies, is highly effective.

Consequently, it seems reasonable to introduce prevention centers with specialized prevention assistants providing long-term care for CAD-patients to achieve sustainable prevention effects, analogous to the concept of heart failure nursing [27]. Obviously, these strategies are more effective than the actual standard of care.

It might be challenging to implement such programmes in various environments; however, involving non-physician health care professionals into long-term risk factor management appears to be an attractive strategy to reduce the work load for physicians, to reduce costs for health care systems and provide better patient care. Health care systems, health insurances and/or pharma groups, who set a focus on prevention, should be convinced to support such strategies.

To optimize efficacy of prevention programmes the optimal duration and selection of patients for the programmes should be evaluated in further studies; these issues are actually investigated by ongoing studies, inter alia from our study group.

Furthermore, it is important to make long-term preventive strategies more attractive to patients to increase adherence. Besides personal contacts by prevention assistants/physicians and besides telemetric control of risk factors, individualization of prevention seems to be a promising way to improve attractiveness of prevention programmes (Figure 3).

Several studies have shown that individual genetic analyses might help tailoring future preventive strategies and might improve long-term results [28,29]. In a meta-analysis of four primary and secondary prevention trials totaling 48,421 individuals, Mega et al. reported that a polygenetic risk score identified patients with increased risk for coronary events; in patients with high polygenetic risk score a greater benefit from statin therapy was observed compared to patients with low genetic risk [28,29]. However, it is unclear if disclosure of genetic risk affects patients’ behaviour or not; it might be argued that disclosure of genetic risk could reduce motivation for lifestyle modifications (“genetic fatalism”). Before genetic risk scores could become widespread tools for individualization of preventive strategies, more studies will be needed [30].

A further promising strategy to individualize prevention is risk assessment by inflammatory biomarkers, such as high-sensitivity C-reactive protein (hsCRP). In the Canakinumab anti-inflammatory thrombosis outcomes study (CANTOS) study, a strong relationship between hsCRP reduction and cardiovascular event reduction following treatment with canakinumab has been shown [31]. It is further known that reduction of hsCRP and inflammatory parameters can be achieved with a healthier lifestyle and increased physical activity [32,33]. It therefore seems reasonable to include inflammatory biomarkers into risk assessment and personalization of cardiovascular prevention.

Another emerging area of research with the potential to lead to personalized preventive treatment is the influence of gut microbiome on the cardiovascular system. Current studies suggest that profiling an individual’s gut microbiome could conceivably guide treatment choice in future [34,35].

The list of research on new biomarkers that might help tailoring management of cardiovascular patients is long. It is becoming obvious that future prevention will have a personalized approach, using information that derive from genetic and biomarker analyses.

## 5. Conclusions

Prognosis of MI-patients is depending on sustainable RFC during long-term course. However, international current “real world” data show that risk factors are not sufficiently controlled in the majority of patients with CAD. An increasing number of trials demonstrated beneficial effects of long-term prevention programmes to improve RFC. The implementation of these programmes into clinical practice and a better public awareness on long-term RFC in patients with CAD have potential to further improve prognosis after MI. To increase attractiveness of long-term prevention programmes innovative strategies, such as telemetric risk factor control and individualization of prevention, should be included.

## Figures and Tables

**Figure 1 jcm-08-01114-f001:**
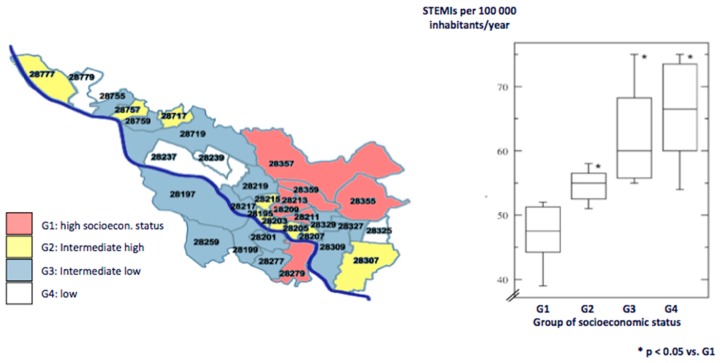
Socioeconomic status and incidence of STEMIs. Postal codes of home addresses of STEMI-patients in the city of Bremen/Germany and a social deprivation index were used to group patients into different categories of socioeconomic status. A negative association between low socioeconomic status and STEMI-incidence was observed with most STEMIs in socially disadvantaged city districts (G3, G4).

**Figure 2 jcm-08-01114-f002:**
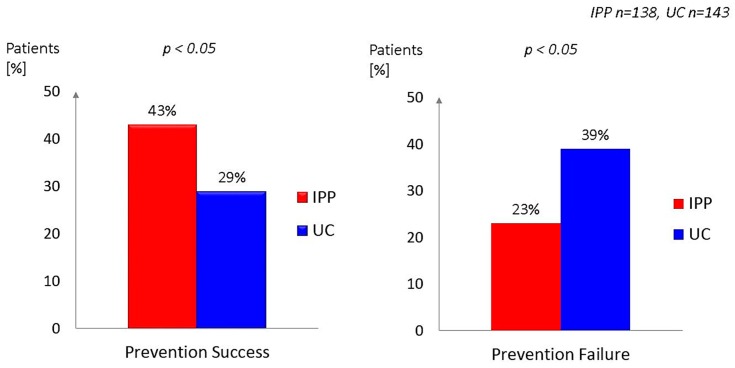
Prevention success and prevention failure associated with 12 months intensive prevention programme (IPP) vs. usual care (UC).

**Figure 3 jcm-08-01114-f003:**
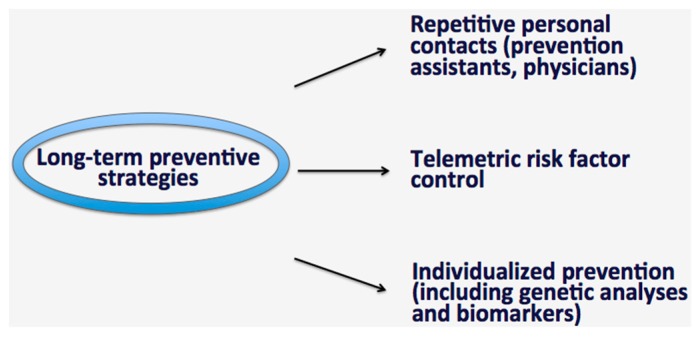
Strategies to increase attractiveness of long-term prevention programmes.

**Table 1 jcm-08-01114-t001:** Definition of the endpoints “prevention success” and “prevention failure” for the IPP study.

**Prevention success** Improvement of one risk factor without deterioration of another **Prevention failure** Deterioration of one risk factor without improvement of another		
**Risk factors**	**Improvement**	**Deterioration**
Smoking	Smoking cessation (at least for 4 weeks at evaluation, controlled by serum cotinine levels)	New or recurrent active smoking
LDL cholesterol	Reduction < 5 mg/dL	Increase ≥ 5 mg/dL
Physical inactivity	Increase of caloric expenditure ≥ 500 kcal/week (leisure time moderate or vigorous physical activity assessed by IPAQ)	Decrease of caloric expenditure ≥ 500 kcal/week

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
