# Peer review of "Long-Term Risk Factor Control After Myocardial Infarction—A Need for Better Prevention Programmes"

_jcm, 2019, doi:10.3390/jcm8081114_

Reviewer 1 Report

The paper is interesting, well prepared and written noticeably. The abstract fits the exact picture of the whole manuscript. The manuscript highlights the importance of cardiovascular risk factors control in secondary prevention. This topic is very important for clinicians.

Minor comments:

Please provide a better definition of Figure 2.

Page 4 In the last paragraph (line 140, 142 and 143) please make a bullet list.

Author Response

Reply to the Review Report (Reviewer 1)

 We would like to thank the reviewer for his/her helpful comments;

in our view the manuscript has improved by the revision.

The revised manuscript has been modified in response to the following comments:

 Please provide a better definition of Figure 2.

Page 4 In the last paragraph please make a bullet list.

 In the revised manuscript a bullet list was added by the inclusion of table 1 that provides better definition of Figure 2 (page 8).

Reviewer 2 Report

The primary aim of the study is to ‘gives an overview on recent health care data regarding RFC in patients after MI and presents data on trials investigating modern preventive strategies to achieve risk factor targets during long-term course.’ So I would think only studies recruited patients with histories of CAD should be reviewed. But in the reference list, there are studies investigated the risk factor associated with the first CVD event (ref 15) and efficacy of statin in high and very-high CVD risk population (ref 14). Further, author aimed to review ‘recent’ health care data regarding RFC, but why studies dated back to 1993 are cited?

 I understand that this is not a systematic review, but is there any mechanism to identify relevant studies for review? is there a potential risk-of-bias in selective reporting?

 In page 4, lines 113 to 118, is it necessary to present the components of the intervention from the primary study? Given the reference is already supplied.

 In page 4, ‘To illustrate the effects of IPP the study endpoints “prevention success” and “prevention failure” are shown in Figure 2. These endpoints were developed in analogy to the RESPONSE II trial [24].’ Since the study RESPONSE II trial is mentioned, why the results from this study are not compared and contrasted in the same figure?

Author Response

 Reply to the Review Report (Reviewer 2)

 We would like to thank the reviewer for his/her helpful comments;

in our view the manuscript has improved by the revision.

The revised manuscript has been modified in response to the following comments:

 1. The primary aim of the study is to “gives an overview on recent health care data regarding RFC in patients after MI and presents data on trials investigating modern preventive strategies to achieve risk factor targets during long-term course.” So I would think only studies recruited patients with histories of CAD should be reviewed. In the reference list, there are studies investigating the risk factor associated with the first CVD event (ref 15) and efficacy of statin in high and very-high CVD risk population (ref 14). Further, author aimed to review ‚recent’ health care data regarding RFC, but why studies dated back to 1993 are cited?

 As suggested by the reviewer the study of März et al. (ref. 14) on statins in a high and very-high CVD risk population and the “old” study of Hambrecht et al. from 1993 (ref. 26) were not cited anymore in the revised manuscript.

 In our view the study of Schmucker et al. (ref. 15) is in a good context to the paper; however, the 5-year follow-up after MI of the study and the context to the concept of life long prevention (according to current guidelines) was now better explained in the revised manuscript (page 5, line 111-119):

 Although acute treatment strategies were comparable between the different groups of socioeconomic status, a 34% increase of MACCE was observed during 5-years course after MI in patients from the most socially deprived city areas compared to those from the city areas with the highest socioeconomic status. These results were pronounced in young STEMI-patients. The data underline the importance of socioeconomic status in cardiovascular prevention. In the context of a lifetime approach to prevention, that is emphasized in the current guidelines, intensified prevention efforts in socioeconomic deprived areas are a promising strategy to improve long-term prognosis after MI.

 2. I understand that this is not a systematic review, but is there any mechanism to identify relevant studies for review? Is there a potential risk-of bias in selective reporting?

 This was added to the revised manuscript on page 4, line 80-82.

Besides own research, a PubMed literature search using the key terms “prevention programme”, “myocardial infarction” and “risk factor control” as well as checking reference lists of journal articles was performed to identify relevant studies for review.

 3. In page 4, lines 113 to 118, is it necessary to present the components of the intervention from the primary study?

 In the revised manuscript the presentation of the intervention has been strongly shortened (page 7, line 148-151).

 4. In page 4, „to illustrate the effects of IPP the study endpoints „prevention success“ and „prevention failure“ are shown in Figure 2. These endpoints were developed in analogy to the RESPONSE II trial [24].“ Since the study RESPONSE II trial is mentioned, why the results from this study are not compared and contrasted in the same figure?

 In the revised manuscript the results of RESPONSE II and IPP were now better compared (page 9, line 173 - page 10, line 191). Due to methodical differences between both studies a comparison in one figure did not seem to be suitable; however, similarities and differences between the studies were now explained more extensively in the revised manuscript:

 Comparing the RESPONSE II and the IPP trial, both studies investigated a modern long-term prevention programme after acute coronary syndromes and evaluated  “prevention success” and “prevention failure” after 12 months. However, in the RESPONSE II trial exclusively lifestyle interventions (such as Weight Watchers®) were performed and the 3 “lifestyle risk factors” body weight, physical inactivity and smoking were defining “prevention success” or “prevention failure”. In contrary, improvement of medical therapy (besides lifestyle interventions) was an important component of the prevention programme in the IPP trial, e.g. advicing patients to increase lipid-lowering medication if LDL cholesterol levels did not meet the recommended targets. Therefore and due to the better evidence of LDL cholesterol reduction compared to weight reduction to improve prognosis after MI, LDL cholesterol instead of weight modification was used to calculate “prevention success” and “prevention failure” in the IPP study.

All three components of „prevention success“ and „prevention failure“ -LDL cholesterol, smoking and physical activity- were significantly improved in the IPP study (while body weight was not significantly reduced). In contrast, in the RESPONSE II study only body weight was significantly reduced, while the risk factors smoking and physical inactivity were not improved (nor was LDL cholesterol).

The comparison of both studies demonstrates that the focus of preventive programmes (e.g. weight reduction or lipid-lowering) has strong implications on achieved prevention results.
